# Flabby Ridge, a Challenge for Making Complete Dentures

**Corina-Laura Ştefănescu** [1,*,†], **Agripina Zaharia** [1,†], **Rodica Maria Murineanu** [1,†], **Cristina Gabriela Puşcaşu** [1,†], **Liliana Sachelarie** [2,*,†] **and Mircea Grigorian** [1,†]

1   Faculty of Dental Medicine, "Ovidius" University Constanta, 124, Bvld. Mamaia, 900527 Constanta, Romania; agrizaharia@yahoo.com (A.Z.); rodicamurineanu@gmail.com (R.M.M.); cristinap@gmb.ro (C.G.P.); mirceagrigorian@yahoo.com (M.G.)

2   Faculty of Dental Medicine, "Apollonia" University Iasi, Pacurari nr.11, 700511 Iasi, Romania

\*   Correspondence: corina.stefanescu@yahoo.com (C.-L.Ş.); lisachero@yahoo.com (L.S.)

†   Authors contributed equally to this work.

**Abstract:** In the case of old mobile denture wearers, one of the main problems is related to the dentures' retention and insufficient stability. Our goal was to improve support and stability using a different type of final impression, with different types of impression materials. In this study we chose a number of three complete edentulous patients who presented for complete oral rehabilitation. They were wearing full acrylic dentures with poor support and stability. Complete examination revealed the presence of the flabby ridge. The impression methods for the prosthetic fields with a flabby ridge differ from the classical method by using techniques that involve the use of at least two impression materials with fluid consistency to record all the details of the prosthetic fields in the final impression, this being done in two steps; the impression of the flabby ridge areas must be done without pressure, in its resting position. The impression methods we applied led to the expected results, and the dentures succeeded in offering the patient the desired functional comfort. Using fluid impression material for the flabby ridge in a resting position, and a fenestrated custom tray, offered a good quality in adaptation, maintenance and stability of the final dentures.

**Keywords:** flabby ridge; impression; complete dentures





## 1. Introduction

The restoration of the dento-maxillary system functions by total prostheses depends on the correct evaluation of the balance between the positive and negative elements of the support areas, the suction areas and the neutral ones of the totally edentulous prosthetic fields. The negative balance of the prosthetic fields represents for the dentist-technician team a moment of great responsibility in choosing the treatment plan. These negative parameters are given by complex interactions of multiple factors such as the intimate adaptation of the mucous-bone support at the base of the denture, its maximum extension, the presence of salivary film, a correct occlusion, the presence of natural tooth antagonists or other pore fitting dentures. Resorption and atrophy of the alveolar ridge are also essential factors that contribute to compromising the retention and stability of mobile prostheses. Examination of patients with old prostheses with increased instability is very important. The presence of the balance mucosa can be easily overlooked in the conditions of prosthetic fields with a negative prosthetic balance. Denture retention can be jeopardized if one of these factors is compromised. Patients wearing mobile dentures require special attention due to compromised oral anatomy, reduced adaptive capacity, systemic disorders and their corresponding medication, a combination of factors that significantly diminish patients' ability to successfully wear mobile prostheses. A flabby ridge occurs by replacing bone with connective fibrous tissue. It is most often found in the anterior part of the jaw, especially when there are remaining anterior mandible teeth or when associated with removable partial dentures in case of l edentulous spacesrepresenting a consequence of ridge overload

and occlusal imbalances [1,2]. The process of resorption and bone atrophy is slow, taking place over a long period of time, not always simultaneously between the mucosa and the bone, the patient adapting to the instability of the dentures, but with consequences on the structure and functionality of the dento-maxillary system. The real problems arise when making new dentures, if the flabby ridge is not properly treated therapeutically. There are several ways in which such a situation can be resolved. The first one is the surgical removal of connective tissue, but with a greatlyreduced bone ridge as a result. From a prosthetic point of view, the difference will have to be compensated by the base of dentures, implying its thickening and the simultaneous increase in weight and volume. Another therapeutic possibility is implant-prosthetic treatment that requires augmentation, implant insertion and overdentures [2]. If the flabby ridge is preserved, a viable therapeutic solution for the success of the mobile treatment is represented by the special impression techniques of the prosthetic field. When the impression is a conventional one, there will always be a degree of compression of the tissues, the natural position of the soft tissues being altered [3]. In the literature, a variety of impression methods have been described for flabby ridges [4–7]. Liddlelow described a technique in which two separate materials are used in a custom tray (using "Paris gypsum" over the area of the flabby ridge and zinc oxide eugenol paste over the rest of the prosthetic field) [8]. Osborne presented an impression method with two different materials as well as two different custom trays used separately for the flabby ridge and the rest of the prosthetic field, subsequently correlating the intraoral results of the two. [9]. Materials and techniques for the treatment of total edentulousness with acrylic dentures have not changed much in the last 45 years [10,11]. Most clinicians use conventional impression techniques, but with the advent of new materials these techniques have been individually adapted to achieve a clearly superior result [12–17]. Now, with CAD-CAM technology for making complete dentures, all these problems related to the impression of flabby mucosa could become history.

The aim of this study is to obtain complete dentures with increased maintenance and stability in the presence of flabby ridges, by applying combined techniques of final impression, using different types of impression materials.

## 2. Materials and Methods

From the subjects who presented, in the Mobile Prosthetic Department, Faculty of Dentistry, "Ovidius" University Constanta, we chose a number of three patients for this study. The selection was made based on the particularities related to the etiology of the appearance of the flabby ridge, the type of flabby tissue, as well as the impossibility of performing an implant-prosthetic treatment due to medical conditions, long treatment time and increased cost.

The first case was chosen to demonstrate the appearance of the flabby ridge due to the combined syndrome in patients who, for various reasons, do not perform complete oral rehabilitation on the two arches, a complete maxillary denture opposing class I lower edentulous ridge.

The second case wasa complex one, the patient having worn complete dentures for over 15 years. The alveolar ridges underwent an advanced process of atrophy and resorption, and the dentures had an advanced stage of infiltration of the acrylic material as well as an accentuated abrasion at the level of the artificial teeth. In order to maintain the complete dentures in the oral cavity, the patient adapted by developing a reverse occlusion with repercussions on the dento-maxillary system.

The third case was chosen due to the rare topography of the flabby ridge, respectively lower fronto-lateral on both hemiarches. The etiology of the appearance of the flabby ridge in this case wasdue to the mastication only with the upper denture on a lower edentulous ridge. The permanent trauma of the mandibular prosthetic field during the exercise of the functions of the dento-maxillary system contributed to the appearance of a flabby tissue on such a large surface.

Clinical examination revealed several aspects: increased instability of complete acrylic dentures wearers (between 5 and 15 years), with major changes in atrophy and resorption of prosthetic fields; infiltrations and color changes of prosthetic parts; almost complete abrasions of artificial teeth, with the impossibility of performing the fundamental functions of dento-maxillary system; modification of the occlusal relations with the appearance of the reverse occlusion or malocclusions due to the attempt to adapt to a compromise situation, the patients trying through new occlusal positions to maintain the dentures during functions.

From our experienceand that of other researchers, the flabby ridge, on the surface of the bone support of complete edentulous prosthetic fields, can have various locations and structures, thus imposing a classification (Figures 1–3):

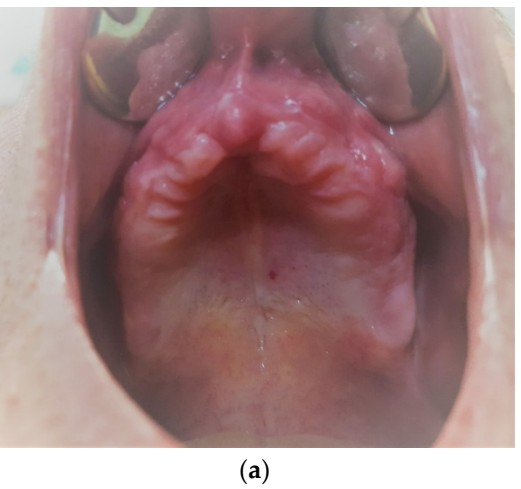

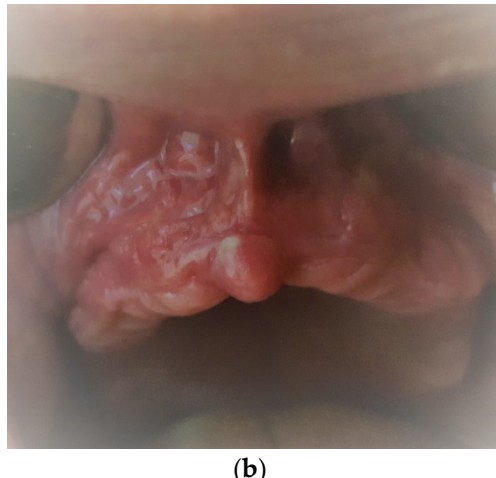

(**a**)    (**b**)

**Figure 1.** (**a**) Fronto-lateral, hypertrophied, hyper keratinized flabby ridge, with high antero-posterior mobility, large thickness, rhomboidal incisive papilla, "drop" appearance, low-insertion labial frenum, deep palatal arch in the anterior portion, neutral tuberosities, ah oblique zone; (**b**) detail of the appearance of the flabby ridge; the arrows indicate the height of the layer of flabby mucosa.

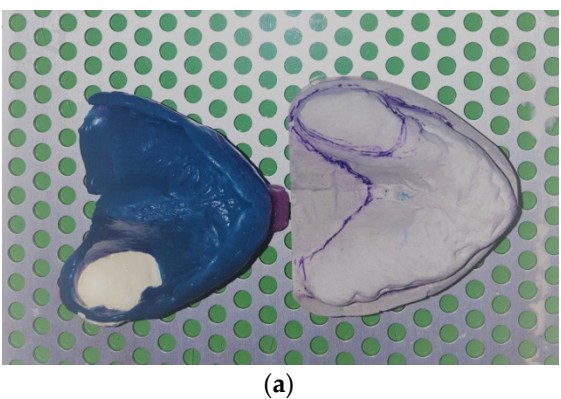

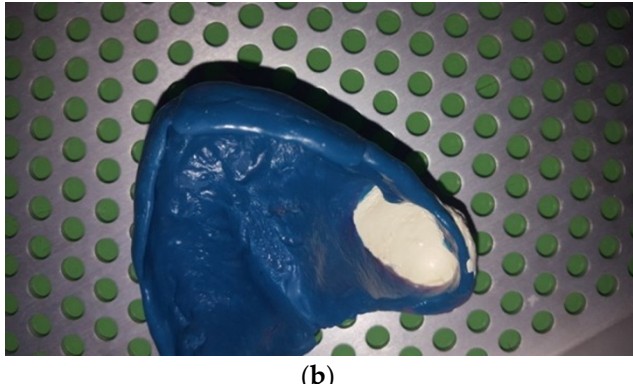

(**a**)    (**b**)

**Figure 2.** (**a**) Preliminary cast and final impression of a V-shaped maxillary prosthetic field, with high retentive ridges, deep palatal vault, retentive tuberosity, with flabby mucosa; (**b**) detail of the upper jaw impression recorded with two low viscosity materials in two stages. The tuberosity impression was made in the resting position, without compression, by brushing the material on its surface.

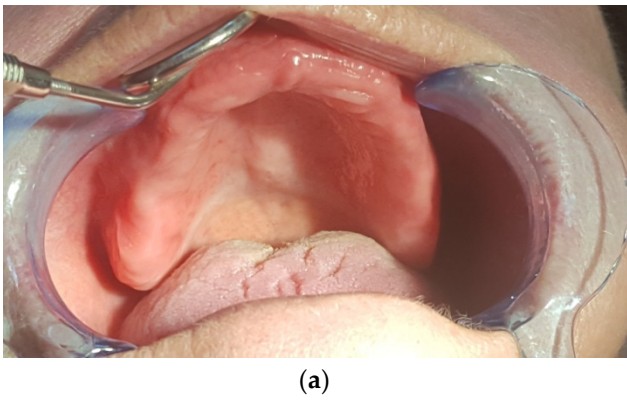
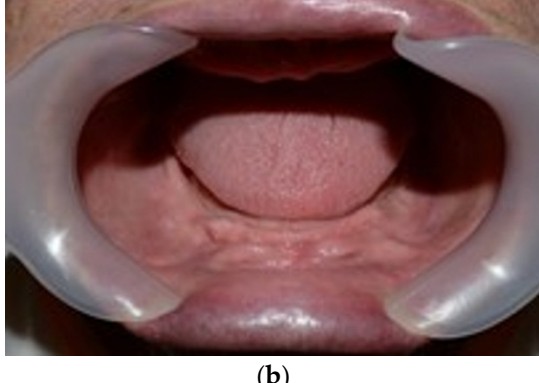

(**a**)　　　　　　　　　　　　　　　　　　　　　　　　　(**b**)

**Figure 3.** (**a**) Maxillary prosthetic field, with well-formed high ridges, voluminous tuberosities, slightly retentive vestibulo-distal, voluminous maxillary torus located posterior, flabby ridge apparently normal histological appearance at inspection, but with increased mobility; (**b**) lower prosthetic field with flabby ridge in the lower frontal area, atrophic, with hyperplasia of the vestibule groove by continuous trauma due to the edge of a lower prosthesis without maintenance and stability.

Depending on location:

- upper frontal edentulous ridge(very often)
- upper fronto-lateral edentulous ridge(often)
- maxillary tuberosity(rare)
- lower frontal edentulous ridge(very often)
- lower fronto-lateral edentulous ridge(rare)
- retromolar pad(often)
- Depending on the structure:
- hyperplasic, keratinized, with high antero-posterior and supero-inferior thickness
- atrophic, thin, small antero-posterior thickness and increased supero-inferior length
- apparently histological normal appearance at inspection, but with increased mobility in all directions.

The preliminary impression was recorded in two of the patients using old dentures as trays, and in the third using standard trays and irreversible hydrocolloid (Tropicalgin-Zhermack, Germany). The custom trays were made of light-cured base material fenestrated in the areas of the flabby mucosa, after the adaptation of the custom tray on the prosthetic field at the level of the peripheral suction areas by the "step by step" method.

Depending on the concept behind the impression techniques, they can be classified into: mucostatic, mucodynamic, with open or closed mouth, compressive or non-compressive, with one or more materials, in one or multiple steps [18]. Our method for final impression was: mucodynamic, two steps, open mouth, non-compressive, compound, by using at least two impression materials.

The final impression was performed experimentally in three ways, using low viscosity impression materials so as not to compress the prosthetic field and moreover to record the flabby mucosa in a resting position. For the first patient, we used: zinc oxideeugenol paste brushed on the flabby mucosa (Repin-Spofa Dental, Jičín, Czech Republic) and fluid condensing silicone (Oranwash-Zhermack, Germany) for the rest of the prosthetic field; for the second patient: zinc oxideeugenol paste brushed for flabby tissue and fluid condensing silicone with superior properties similar to addition silicones (Speedex-ColteneWhaledent), and for the third, fluid self-mixing addition silicone (ELITE HD +, super light body, Zhermack, Germany) by injection for the flabby tissue and for the rest of the prosthetic field in two steps.

## 3. Results

### 3.1. CASE REPORT 1

A 59-year-old patient presented to the Dental Prosthetics Clinic for prosthetic rehabilitation of the dento-maxillary system. She had been wearing complete acrylic dentures for

over 10 years. At the complete examination, corroborated with the data received from the general practitioner treating the patient's ailments, we analyzed the following important data for the future treatment plan: the superior denture wasunstable on the prosthetic field, infiltrated and unhygienic (Figure 4). The maxillary prosthetic field on inspection seemed to have a positive balance, but palpation proves the presence of the flabby mucosa in the frontal area. Due to the lack of hygiene and the untimely wearing of a denture with low maintenance and stability, the appearance of the field washyperemic, presenting erosive lesions certifying the diagnosis of stomatitis under the prosthetic plate. (Figure 5a,b). At the mandible level, there wasa Kennedy class I edentation, not treated.

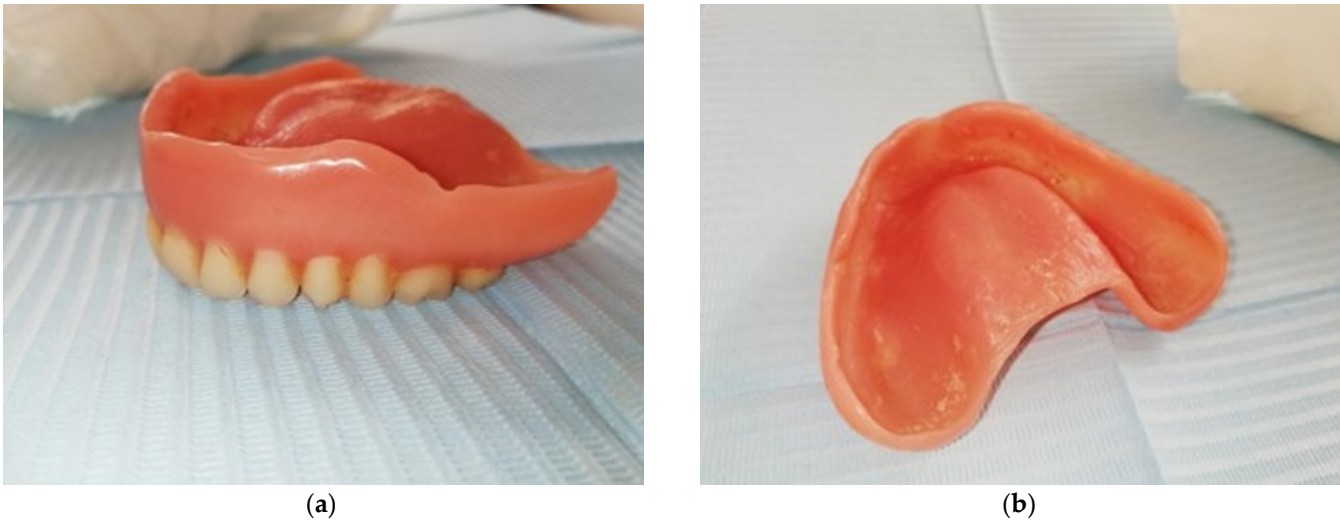

(**a**)                                                                                              (**b**)

**Figure 4.** Maxillary denture about 10 years old (**a**) lateral view (**b**) mucosal view.

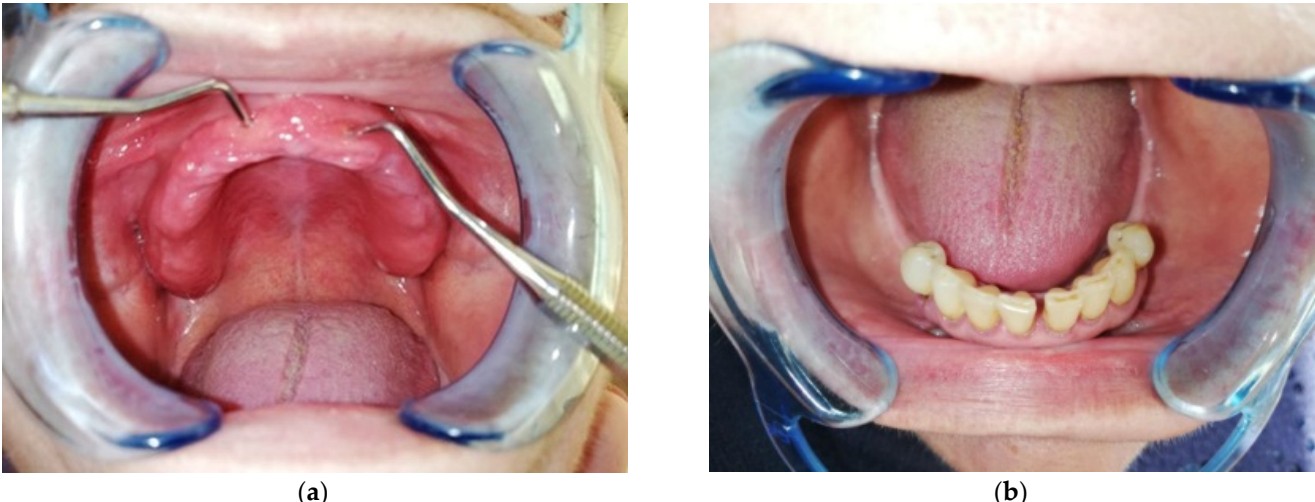

(**a**)                                                                                              (**b**)

**Figure 5.** Prosthetic field examination (**a**) maxillary with frontalflabby ridge; (**b**) mandible Kennedy class I edentation.

The treatment plan excluded surgeries to aid prosthetic treatment and included: local and general anti-inflammatory, antifungal treatment to restore the physiological appearance of the upper prosthetic field and making a new maxillary complete denture and a lower flexible partial denture. The upper preliminary impression was recorded using the patient's old denture as a tray holder, which was disinfected and a layer of approximately 1 mm was removed from the mucosal surface and edges. The impression material was fluid condensing silicone (Oranwash-Zhermack, Germany). (Figure 6a). The

custom tray, made of light-cured base material, was windowed over the entire surface of the flabby area. (Figure 6b).

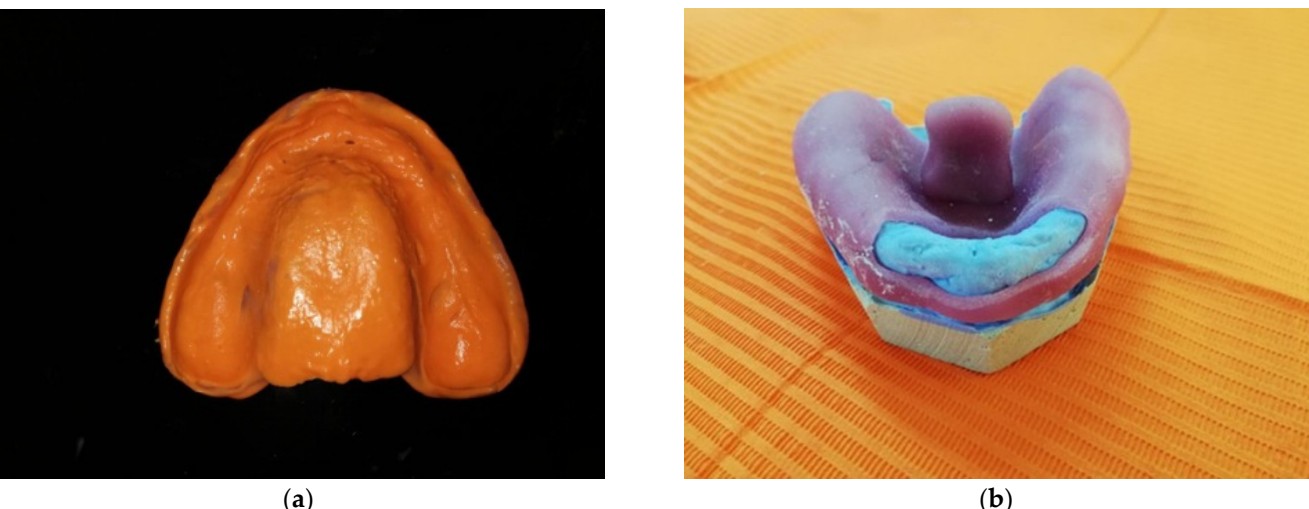

**(a)**　　　　　　　　　　　　　**(b)**

**Figure 6.** (**a**) Preliminary impression; (**b**) Fenestrated custom tray.

After custom tray adaptation (Figure 7a) the final impression of the prosthetic field without the flabby ridge area was recorded. Excess material from this area was removed with a scalpel (Figure 7b).

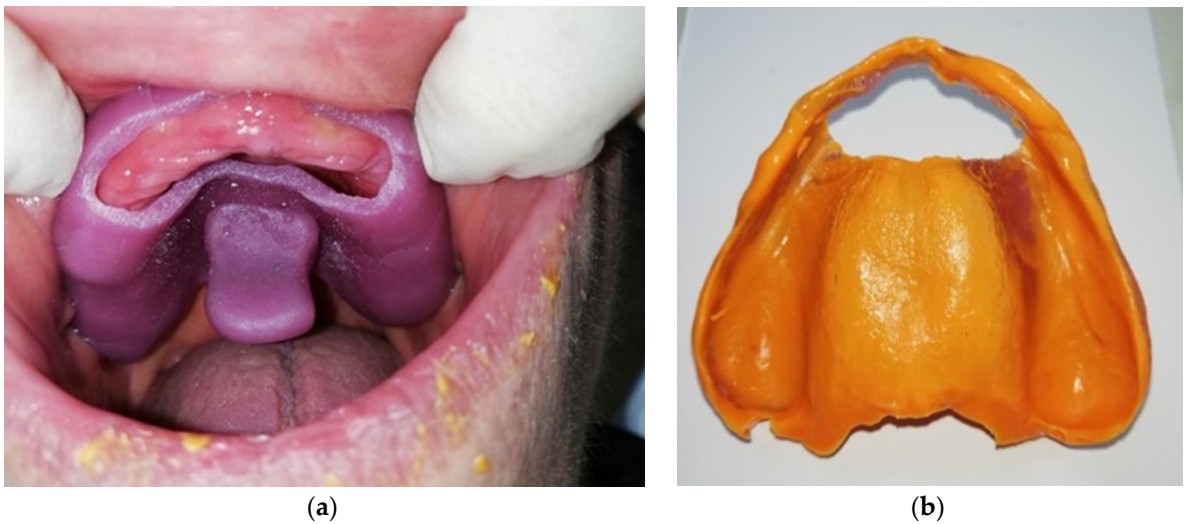

**(a)**　　　　　　　　　　　　　**(b)**

**Figure 7.** (**a**) Custom tray adaptation; (**b**) first step of final impression after removing the material from the flabby ridge area.

The impression of the flabby mucosa area was performed by brushing the zinc oxide eugenol paste on its surface, in its resting position. (Figure 8a,b).

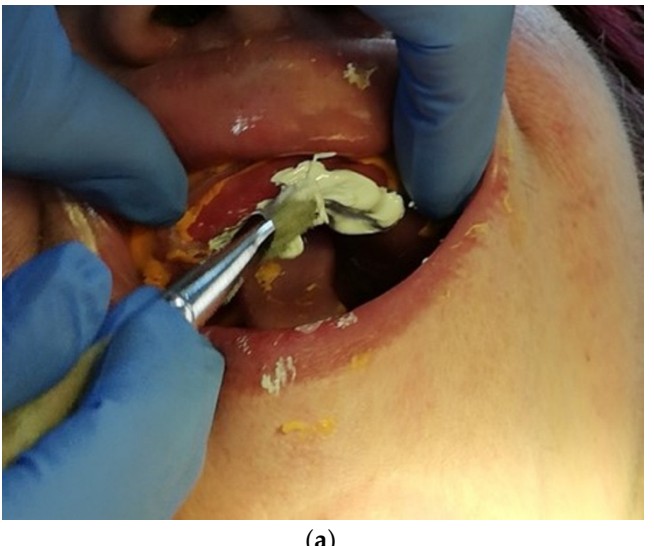
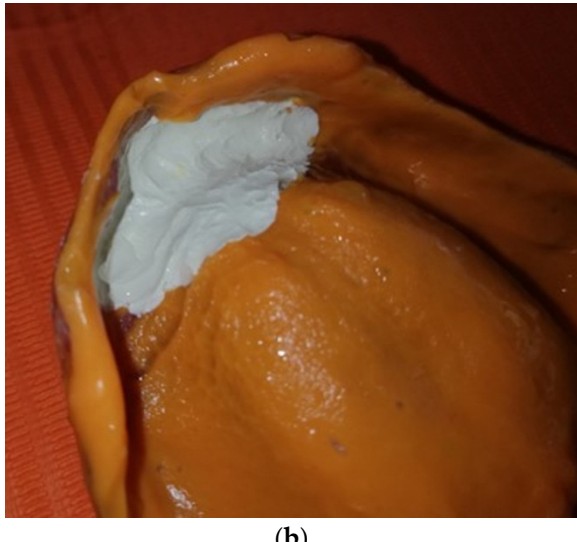

(**a**)  (**b**)

**Figure 8.** (**a**) flabby mucosa impression technique; (**b**) details of final impression.

After establishing occlusal relationships and wax dentures try-in occlusal adaptation, the maxillary complete denture and a biodentaplast partial flexible mandible denture were applied in the oral cavity, their maintenance and stability being of good quality. (Figure 9a,b).

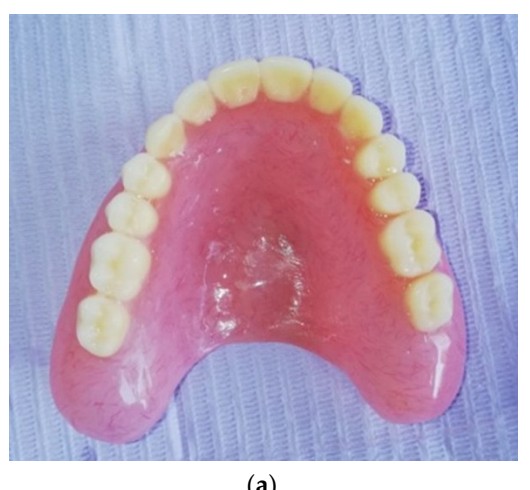
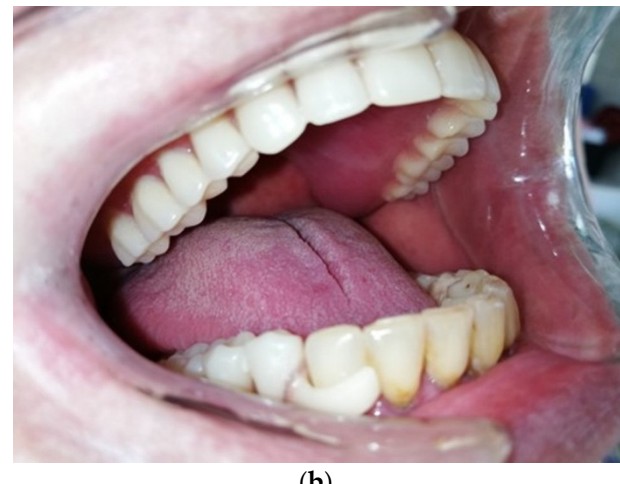

(**a**)  (**b**)

**Figure 9.** Final denture (**a**) extra-oral view (**b**) intra-oral view.

*3.2. CASE REPORT 2*

A 75-year-old patient presented to the Dental Prosthetics Clinic for the morpho-functional restoration of the dento-maxillary system. She had been wearing complete acrylic dentures for about 15 years. After complete dental examination corroborated with the data received from the general practitioner who treated the patient's ailments, we couldanalyze the following important data from the point of view of the future treatment plan: the patient had a series of chronic general ailments that required caution in performing orofacial surgical treatments; the dentures were old, unstable, with a high degree of abrasion and infiltration of the acrylic material, reasons for which the patient changed her occlusion (mandible position is protruded resulting a reversed occlusion) (Figure 10a); the maxillary prosthetic field had more elements of negative than positive prosthetic balance; the mandibular prosthetic field had a flabby ridge on the entire frontal area, mucosa of the vestibular frontal sulcus had undergone a process of hyperplasia, the edentulous

lateral mandibular ridge wasatrophied, approaching, according to the classifications in the literature, a fifthgrade ridge [19] (Figure 10b). The treatment plan excluded any kind of surgery that could help the prosthetic treatment due to the general diseases and conditions that imposed restrictions. Consequently, pre-prosthetic treatment began by conditioning the tissues of the prosthetic fields by removing areas of the denture that affected the mucosal structures, restricting the wearing of it except during food periods, and a local and general anti-inflammatory treatment.

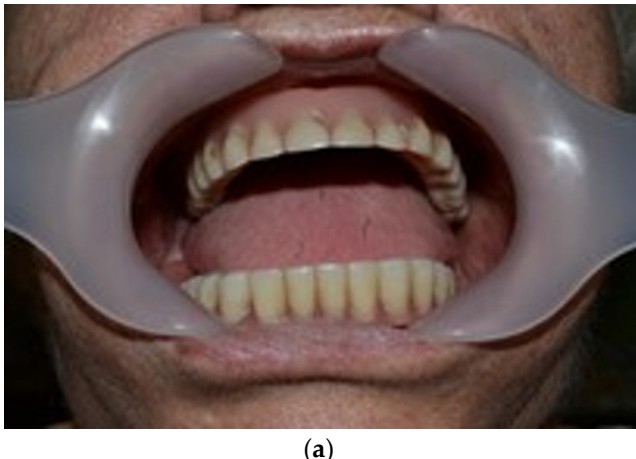
(**a**)
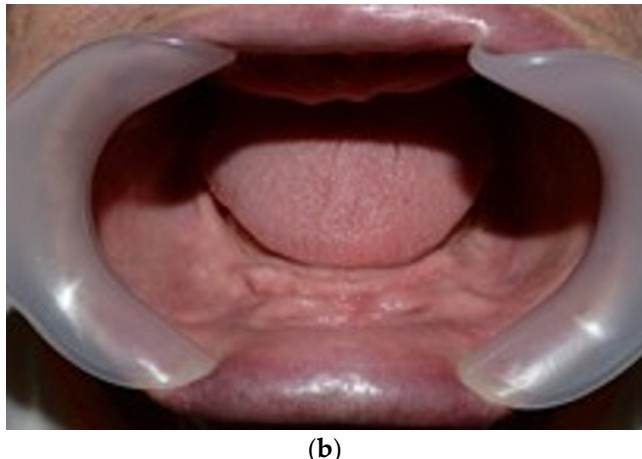
(**b**)

**Figure 10.** (**a**) protruded mandible position due to unstable and high abrasion of the artificial teeth; (**b**) appearance of the mandible prosthetic field with frontal flabby ridge.

In this case we chose the method of fenestrating the custom mandibular tray and also the same impression technique as in the previous case, respectively the brushed zinc oxide eugenol paste, but for recording the rest of the prosthetic field we used a condensing silicone with superior humidity control properties, due to increased salivary flow of our patient (Speedex-ColteneWhaledent) (Figure 11a,b).

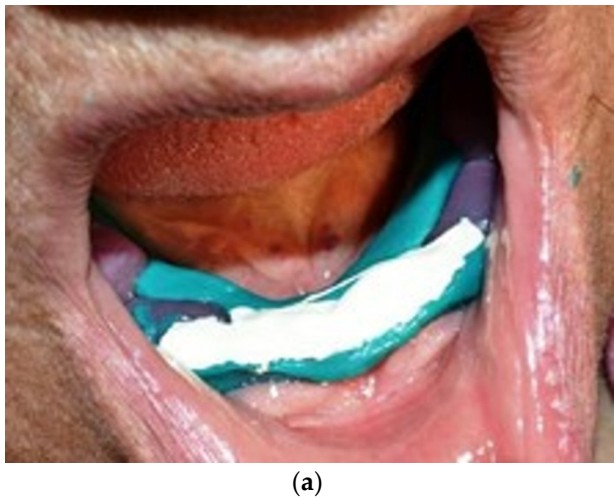
(**a**)
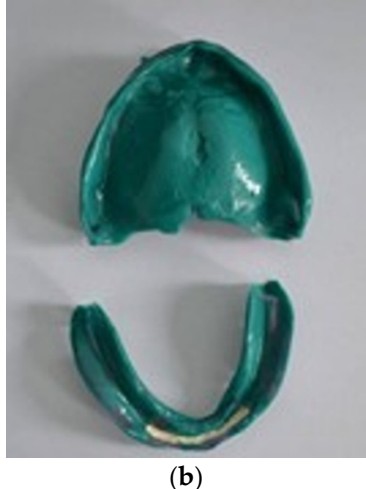
(**b**)

**Figure 11.** Final impressions (**a**) The appearance of the final mandible impression; (**b**) Appearance of the two final impressions.

The final result improved dentures maintenance and stability, restoration of occlusion, and a special esthetic (Figure 12).

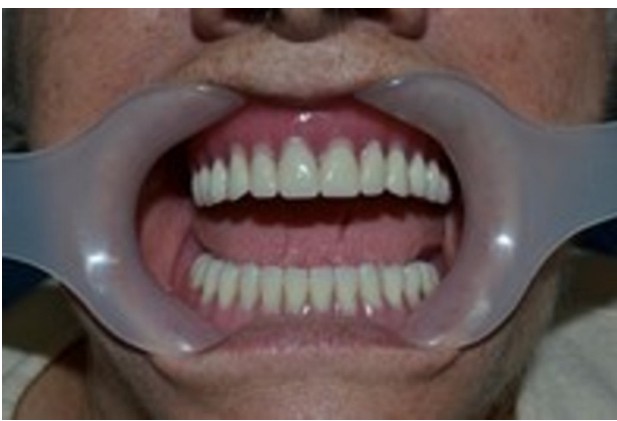

**Figure 12.** Final dentures in the oral cavity.

*3.3. CASE REPORT 3*

A patient, aged 65 years, complete dentures wearer for about 5 years, presented to the Dental Prosthetics Clinic for the restoration of the lower denture. The patient had not used the lower denture for the past two years due to increased instability. Examination revealed the presence of the inferior fronto-lateral flabby ridge on both sides of the arch (Figure 13a). The treatment plan included: making a complete mandible acrylic denture, with a classic preliminary impression in a standard tray and alginate. The custom tray was fenestrated over the entire surface of the flabby ridge area (Figure 13b). The final impression was also performed in two stages: initially the entire prosthetic field and then the two areas of the flabby tissue. The material used was an injection-applied self-adhesive silicone (Elite HD +, super light body, Zhermack, Germany) (Figure 14a,b).

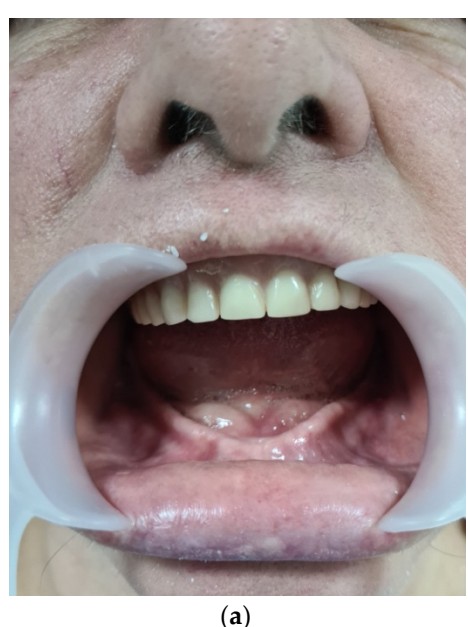
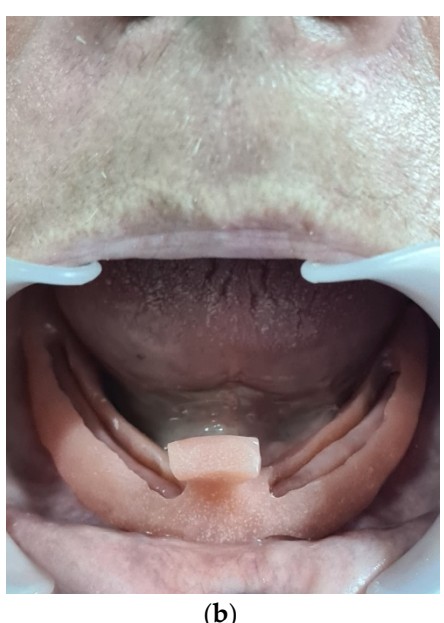

(**a**)                                                                                     (**b**)

**Figure 13.** (**a**) fronto-lateral mandible flabby mucosa; (**b**) fenestrate custom tray.

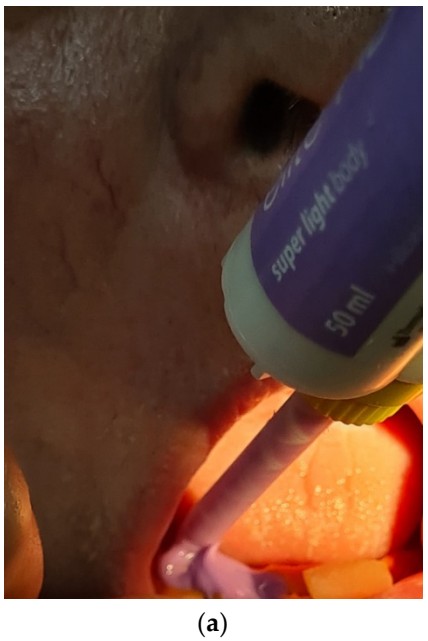

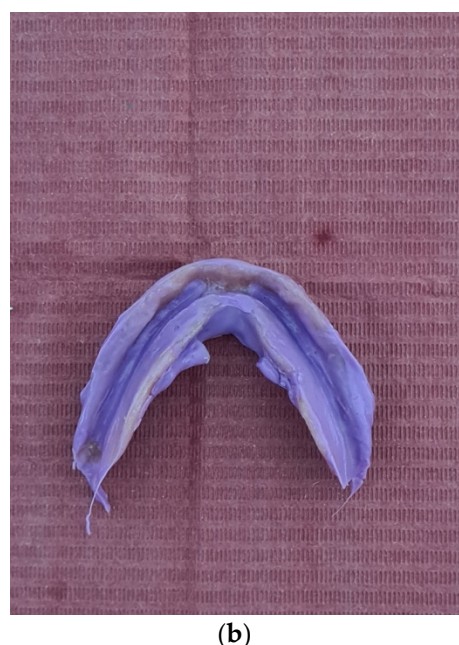

(**a**)                                                            (**b**)

**Figure 14.** (**a**) applying the impression material on the flabby mucosa; (**b**) final lower impression.

Following the determination of the occlusal relations, the balanced occlusion was performed. The final denture has a good suction, it is esthetic, the adaptation of the patient being made easily. The final result is a correct one, the patient being satisfied with the restoration of the mastication, esthetic and phonetic functions (Figures 15 and 16).

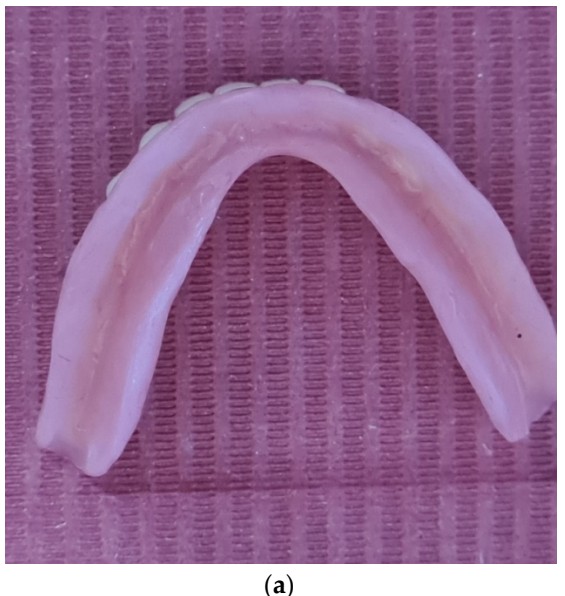

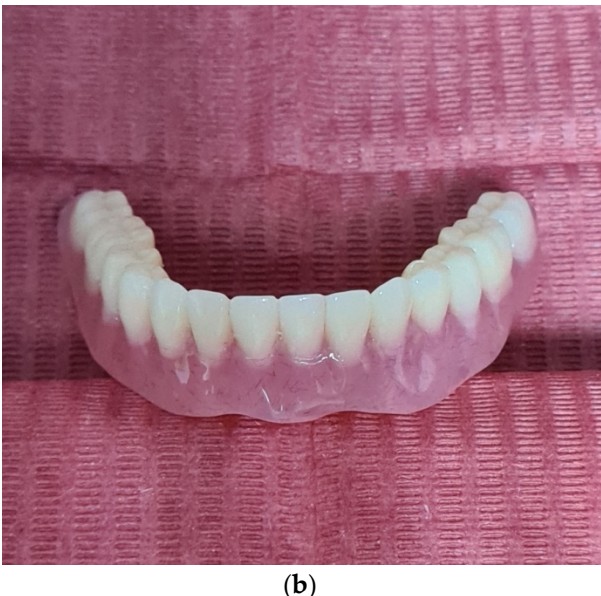

(**a**)                                                            (**b**)

**Figure 15.** Final lower denture (**a**) mucosal view; (**b**) external view.

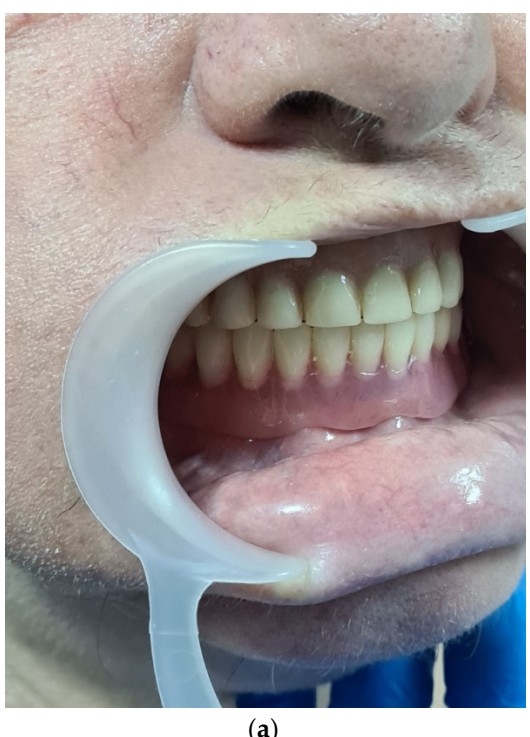 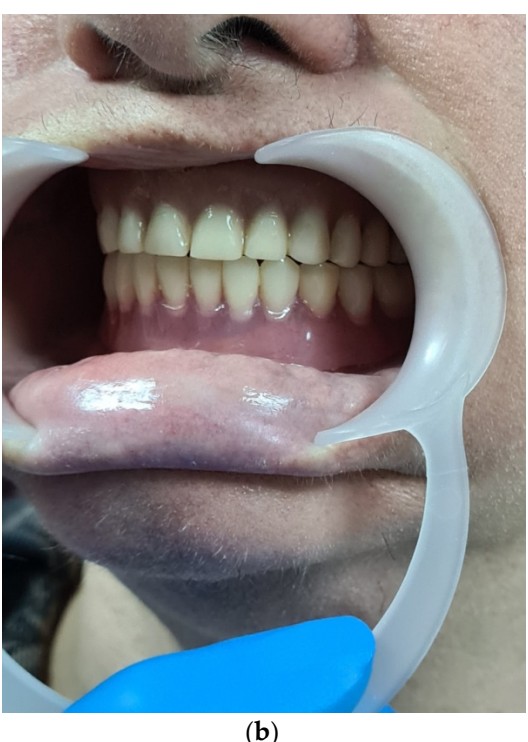

(**a**)            (**b**)

**Figure 16.** Intra-oral final denture (**a**) left view; (**b**) right view.

## 4. Discussion

Edentulous prosthetic field impressions can be a challenge when the residual ridges present less ideal conditions, especially when there is a minimum bone height, and a non-adherent mucosa, mobile in all directions. Geriatric patients, in most of the cases, due to chronic medical conditions or medical treatments are inadequate for surgical procedures to excise the mobile mucosa, increase bone ridges, sinus lift and apply dental implants. As a result, the management of the negative elements of the prosthetic field can be achieved by extending the basic principles of making complete dentures, without resorting to invasive procedures. Particular or individualized impression techniques, in the case of flabby ridges, do not involve additional knowledge, but only fundamental knowledge of the biodynamics of complete dentures. The impression technique is easy and uses materials that the practitioner is already familiar with. Another important element is to obtain a stable, balanced occlusion, without which the maintenance and stability of the prostheses cannot be obtained even if the adhesion and suction by impression have been obtained. Over time, various types of approach and management of this type of mucosa have been analyzed and experienced, which is a challenge for any practitioner [20–24]. Devlin H (1985), Linch C.D. (2003), describe similar methods of managing the flabby ridge, indicating in their studies a certain imprinting protocol, namely: the technological realization of the fenestrated custom trays; a pressure-controlled impression technique; use of several fluid impression materials and the combined use of mucostatic and mucodynamic techniques [25,26]. Other authors have chosen another impression method for the mandible flabby ridge using the tunneling method. After marking the areas of interest on the surface of the cast, wax spacers of the same size were used, the custom tray being made at a distance and covered with a uniform layer of wax in the corresponding areas, the impression being made by injecting a plaster paste [27]. The technique that uses two-step impression, as well as the mucostatic technique, are considered appropriate by most authors.

Other authors are using perforated custom trays made at distance from the flabby tissueand the final impression is made in one step with a single material polyvinilsiloxane or addition silicon [28,29]. An interesting method of making a complete maxillary denture with a flabby ridge in the frontal area was developed by Nawaf Labban (2018). The tech-

nique consisted of performing a pressure relief by means of a custom tray not windowed, but made at a distance from the prosthetic field, and a polyvinylsiloxane monophase impression material [30].

## 5. Conclusions

Examination of old dentures wearers with increased instability is very important. If the causality of this type of mucosa is overlooked and its impression is not done properly, the new dentures will have the same instability as the old ones. The impression of flabby ridge must be done without compression in its resting position using specially made custom trays. The materials used must have high fidelity to the details, low viscosity and high adhesion to the tissues.

**Author Contributions:** Conceptualization, C.-L.Ş.; methodology, C.-L.Ş., A.Z.; R.M.M., software, M.G., C.G.P., validation, R.M.M., C.G.P.; resources, A.Z., writing—C.-L.Ş.; visualization, R.M.M.; supervision, C.-L.Ş., A.Z., L.S., methodology, visualization, supervision. All authors have read and agreed to the published version of the manuscript.

**Funding:** This research received no external funding.

**Institutional Review Board Statement:** All subjects gave their informed consent for inclusion before they participated in the study. The study was conducted in accordance with the Declaration of Helsinki, and the protocol was approved by the Ethics Committee of "Ovidius" University of Constanta, Romania (169/11/01/2017).

**Informed Consent Statement:** Informed consent was obtained from all subjects involved in the study.

**Data Availability Statement:** Not applicable.

**Conflicts of Interest:** The authors declare no conflict of interest.

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
