# Peer review of "Flabby Ridge, a Challenge for Making Complete Dentures"

_applsci, doi:10.3390/app11167386_

Round 1
Reviewer 1 Report
The number and quality of references are low. Only 17 references and most of them are very old. It is mandatory to put new references.
The quality of the photos is low, making it difficult to see the problem and the results. Better photos can help readers to see the problem and the solution.
Nothing new, only the material used in the open window in the tray is changed. Some digital approaches would be more appropriate due to the present of prosthodontics.
Author Response
The authors acknowledge the useful observations and suggestions of the reviewer’s as concerns the manuscript entitled : Flabby Ridge, a Challenge for Making Complete Dentures, Corina-Laura Åžtefănescu 1*, Agripina Zaharia 1, Rodica-Maria Murineanu 1, Cristina-Gabriela PuÅŸcaÅŸu 1, Liliana Sachelarie2, Mircea Grigorian 1
According to the reviewer’s recommendations, all the suggestions were taken into account.
We will attach the new version of the article with all the modifications that you asked for:
The new version of the article has English language spelling and grammar checked.
We updated the references with new articles and case reports.
We changed the resolution (300 dpi) and size of the pictures.
Regarding the novelty of our research, from the study of the bibliography, most authors use for the impression of flabby ridge custom trays at distance, perforated, and the technique itself varies from one step impression, with a single material, to two steps impression by injecting the plaster of Paris or a polyvinyl siloxane on the area of the flabby ridge. Most impressions are mucostatic. We chose the fenestrated custom tray. The final impression method is mucodynamic, in two stages. In the first two clinical cases, zinc-oxide-eugenol paste was used by the method of brushing on the flabby tissue, a method that we consider optimal due to the properties and advantages of this material: increased fidelity of details, fast setting. In the third case we used a quick-setting, high-fidelity addition silicone, so that the material does not flow and there are no unmarked spaces or excess material in the area of flabby ridge
We also classified the balancing mucous membranes according to topography and structure, data that we did not find anywhere in the references and literature studied.Regarding the complete dentures made by the digital method, although the optical impression of the prosthetic field, or the scan of the preliminary cast registers a natural, resting position of the balance mucosa, no comparative studies have been performed between classic and digital prosthesis. Digital dentures have a number of advantages, but also disadvantages. Moreover, not every patient can benefit from dentures made by CAD-CAM technology, due to their socio-economic situation, the cost being high.
I remain most respectfully yours,
Prof.dr.Liliana Sachelarie
Reviewer 2 Report
The article is well written and interesting
Author Response
Thank you, for the report review
We will attach the new version of the article with the modifications that you asked for:
The new version of the article has English language spelling and grammar checked.
I remain most respectfully yours,
Prof.dr.Liliana Sachelarie
Reviewer 3 Report
The paper presents an interesting methodology of managing flabby ridges for edentulous patients. However, the overall flow of the paper is hampered by difficulties with the use of English in the paper. Tenses, plurals, appropriate wording in description made a potentially relevant paper difficult to read and follow the thought process of the authors.
The overall flow of the paper is sputtered in its flow. A rewrite and improvement of the thinking on its presentation can make a significant improvement to what could be a valuable paper.
Author Response
The authors acknowledge the useful observations and suggestions of the reviewer’s as concerns the manuscript entitled : Flabby Ridge, a Challenge for Making Complete Dentures, Corina-Laura Åžtefănescu 1*, Agripina Zaharia 1, Rodica-Maria Murineanu 1, Cristina-Gabriela PuÅŸcaÅŸu 1, Liliana Sachelarie2, Mircea Grigorian 1
According to the reviewer’s recommendations, all the suggestions were taken into account.
We will attach the new version of the article with all the modifications that you asked for:
The new version of the article has English language spelling and grammar checked.
We made several modifications to improve the introduction, material and methods, results and also an increases number of new references.
I remain most respectfully yours,
Prof.dr.Liliana Sachelarie
Reviewer 4 Report
Dear Authors,
I would like to thank you fot the hard work and dedication to this project and for the opportunity to review this article. This work is very interesting and well structured.
I have few questions that I would like to have explained.
Materials and Methods:
- Please specify at the beginning of material and methods, the type and number of subjects in the sample and how they were chosen; in the abstract you say there were three patients, why you choose this sample? Did you exclude any other patients?
- Please, specify also the location where the study was conducted?
- Was there any Ethical Committee approval?
- The majority of the references are really dated, please try to update them if possible.
- You presented the work as an 'Article', but the core part is described as 'Case Reports'. Shouldn't it be presented as that?
Author Response
The authors acknowledge the useful observations and suggestions of the reviewer’s as concerns the manuscript entitled : Flabby Ridge, a Challenge for Making Complete Dentures, Corina-Laura Åžtefănescu 1*, Agripina Zaharia 1, Rodica-Maria Murineanu 1, Cristina-Gabriela PuÅŸcaÅŸu 1, Liliana Sachelarie2, Mircea Grigorian 1
According to the reviewer’s recommendations, all the suggestions were taken into account.
We will attach the new version of the article with all the modifications that you asked for:
The new version of the article has English language spelling and grammar checked.
We specified in the new vertion of the article that:
From the subjects who presented, in the Mobile Prosthetic Department, Faculty of Dentistry, “Ovidius” University Constanta, we chose a number of three patients for this study. The selection was made based on the particularities related to the etiology of the appearance of the
flabby ridge, the type of flabby tissue, as well as the impossibility of performing an implant-prosthetic treatment due to medical conditions, long treatment time and increased cost.
The first case was chosen to demonstrate the appearance of the flabby ridge due to the combined syndrome in patients who for various reasons do not perform complete oral rehabilitation on the two arches, a complete maxillary denture opposing class I lower edentulous ridge.
The second case is a complex one, the patient being wearing complete dentures for over 15 years. The alveolar ridges underwent an advanced process of atrophy and resorption, the dentures had an advanced stage of infiltration of the acrylic material as well as an accentuated abrasion at the level of the artificial teeth. In order to maintain the complete dentures in the oral cavity, the patient adapts by developing a reverse occlusion with repercussions on the dento-maxillary system.
The third case was chosen due to the rare topography of the flabby ridge, respectively lower fronto-lateral on both hemiarches. The etiology of the appearance of the flabby ridge in this case is due to the mastication only with the upper denture on a lower edentulous ridge. The permanent trauma of the mandibular prosthetic field during the exercise of the functions of the dento-maxillary system contributed to the appearance of a flabby tissue on such a large surface.
We have the Ethical Committee approval: protocol was approved by the Ethics Committee of “Ovidius” University of Constanta, Romania (169/ 11.01.2017)
We updated the references with new articles and case reports.
I remain most respectfully yours,
Prof.dr.Liliana Sachelarie
Round 2
Reviewer 1 Report
The article has been improved in general.
The images are not of sufficient quality for the article, but it is impossible to improve and/or repeat them. Therefore, I can accept them for the article.
Author Response
Thank you for the report review. According to the reviewer’s recommendations, all the suggestions were taken into account.
I remain most respectfully yours,
Prof.dr. Liliana Sachelarie
Reviewer 3 Report
Please see attached document with further comments and requests for improvement to the paper.

Author Response
Dear reviewer,
Thank you for receiving our manuscript and considering it for review. We appreciate your time.
According to the recommendations, all the suggestions were taken into account, as follows:
- In Introduction the requested changes were made, the sentences were rewritten.
- The required phrase was introduced in the Discussions (L284-L285)
- the L310 phrase was taken from the conclusions
I remain most respectfully yours,
Prof.dr. Liliana Sachelarie